# How can past sea level be evaluated from traces of anthropogenic layers in ancient saltpans?

Benny Bechor[1]*, Simona Avnaim-Katav[2], Steffen Mischke[3], Slobodan Miko[4], Ozren Hasan[4], Maja Grisonic[5], Irena Radić Rossi[5], Barak Herut[2,6], Nimer Taha[6], Naomi Porat[7], Dorit Sivan[1]

1 Maritime Civilizations Department, School of Archaeology and Maritime Cultures, University of Haifa, Haifa, Israel, 2 Israel Oceanographic and Limnological Research (IOLR), Haifa, Israel, 3 Institute of Earth Sciences, University of Iceland, Reykjavík, Iceland, 4 Croatian Geological Survey (HGI), Zagreb, Croatia, 5 Department of Archaeology, University of Zadar, Zadar, Croatia, 6 Department of Marine Geosciences, L. Charney School of Marine Sciences, University of Haifa, Haifa, Israel, 7 Geological Survey of Israel (GSI), Jerusalem, Israel

* bbechor1@gmail.com

**Data Availability Statement:** All relevant data are within the paper and its Supporting Information files.

## Abstract

Footprints of human activities identified in the sedimentary sequence of submerged historical saltpans can reveal the history of the site and can indicate the relative sea level during its operational period. Saltpans are man-made constructions used continuously for salt production in the Mediterranean at least for the last 2000 years. The east Adriatic coast contains many such submerged remains, preserved and well-dated by historical archives. Sedimentological, microfossil and geochemical analyses of the sediments from cores drilled in the saltwork area at Brbinj, Dugi Otok, Croatia, enable the reconstruction of various past environmental conditions. The current study aims to: a) identify the anthropogenic unit in the sedimentary sequence deposited over time, b) determine its age, and c) use it as past sea-level limiting points. Basal units made of *terra rossa* soil materials were identified in the sedimentary records. These layers are located -120 ±7 cm below mean sea level next to the separation wall and -125 ±7 cm and -135 ±7 cm, respectively, in the inner pools, most likely representing a man-made pavement. The *terra rossa* layer is overlaid by a unit rich in faunal remains dominated by euryhaline foraminifera and ostracod species such as *Ammonia veneta* and *Cyprideis torosa*, representing the saltworks unit. The flooding of the saltpans by the rising sea is manifested by the deposition of an upper sedimentary unit dominated by remains of marine species. The base and the top of the saltwork unit are dated by Optically Stimulated Luminescence to 1040±50 CE and to 1390±30 CE, respectively. The study presents a new approach for obtaining footprints of human activities in ancient, submerged saltpans, by identifying and dating the indicative anthropogenic layers and using these for the reconstruction of paleo sea-level. The described method can be applied all around the Mediterranean.

**Funding:** The author(s) received no specific funding for this work.

**Competing interests:** The authors have declared that no competing interests exist.

# 1 Introduction

Saltpans are intertidal facilities for salt production present worldwide along coastal areas, including those of the Mediterranean since antiquity. Salt production from seawater by solar and wind evaporation in intertidal facilities is a traditional process used for at least the last two millennia [1–3]. During the operational season (from May to September along the Adriatic coasts), high-tide seawater is channeled into a system of shallow basins through artificial trenches and sluice gates. The seawater flows through the evaporation basins following a low gradient and subsequently reaches higher brine concentrations up to the crystallization stage when salt is produced and can be harvested [4, 5].

Coring in marginal marine environments and archaeological sites in the Mediterranean basin, such as wetlands, lagoons and salt marshes, is a common practice to reconstruct and date paleo sea level and paleo shorelines [6–8]. Faivre et al. [9], Marriner et al. [10], Felja et al. [11], Shaw et al. [12] and Brunović et al. [13] studied the eastern Adriatic coast while Barbero et al. [14], Donnici and Barbero [15] and Fontana et al. [16] studied the lagoon of Venice in the northern Adriatic. The saltpans resemble enclosed lagoons, but their sedimentary and pale-ontological evolution through time and their use as a tool for reconstructing past relative sea level (RSL) was not investigated so far.

Coastal ecosystems include benthic foraminifera (amoeboid protists) and ostracods (micro-crustaceans) as the most abundant and important organism groups. These organisms which partly produce calcareous skeletal remains, depend on various ecological conditions such as substrate type, dissolved oxygen concentration, food availability and salinity [17–22]. In shallow aquatic environments, these parameters are often closely related to water depth and sea level [23, 24]. Foraminifera and ostracods demonstrate great sensitivity to environmental change that might be expressed by variations in the assemblage composition and distribution patterns, for example, by disappearance or appearance of species and/or replacement by other taxa; shifts in abundances and diversity; and abnormalities in test features [18, 20–22, 25, 26]. Because of their pervasive distribution, adaptability to various ecological conditions and excellent preservation in the sedimentary records, the remains of these microorganisms have been widely used as key proxies for deciphering past environmental conditions. Linkages between foraminiferal and ostracod assemblages and related environmental conditions were already discussed by Avnaim-Katav et al. [20–22] for the eastern Mediterranean coast, by Meriç et al. [27] and Barut et al. [28] for the Aegean Sea, and by Brunović et al. [13] for the northern Adriatic coast, supported also by geochemical and sedimentological analyses of the sediments. Sedimentary sequences also represent detailed paleoenvironmental archives accessible through element-concentration data. Geochemical analysis of sediment cores enabled the identification of elements of terrigenous origin, such as Sr, Fe, K, Ti, Mg and Al, which differ from elements that are typical to the marine environments. The relative increase of representative terrigenous elements vs. carbonate can indicate the accelerated deposition of terrigenous sediments [29, 30].

RSL is a combination of an eustatic component (ice-equivalent volume), an isostatic component (vertical readjustment of the crust associated with ice and water load), and tectonic contributions (vertical displacement of the crust associated with continued and gradual vertical movement over time or sudden tectonic activity). Over the last two millennia, the dominant component of the glacial isostatic adjustment (GIA) in the Adriatic is the hydro-isostasy, driven by global melting, but the northern coast may also have been affected by the glacioisostasy due to its proximity to the Alps and to the location of the former Scandinavian continental ice sheet [7, 31, 32].

RSL data along the Dalmatian coast for the last millennium are insufficient and not consistent. Biological RSL proxies examined by Faivre et al. [33] at the islands of Vis and Biševo in

central Dalmatia, dated to the medieval period, provide RSL data which spread over a large vertical distance ranging from 15 to 150 cm below the current mean sea level (MSL). Sedimentological indicators assessed by Marriner et al. [10] from Caska on the island of Pag, suggested RSL that range between 60 to 90 cm below the current MSL for the period between the 11[th] - 13[th] centuries. Shaw et al. [34] indicated RSL rise since the early 18[th] century of about 28 cm using salt-marsh reconstructions in central Dalmatia. Brunović et al. [13] inferred RSL rise at Cres Island through the Holocene from analyses of sediment cores from coastal karst dolines, dating the flooding of the paleo-marine pond (Arcij) to around 1240 CE and estimating its depth at 50 cm below the current MSL. Recently, Bechor et al. [35] investigated four saltpans in central Dalmatia (including the Brbinj saltpans) as archaeological sea-level indicators by continuous, high-resolution elevation measurements of the separation walls and the sluice gates. Their study suggested that the top of the separation wall of the saltpans was originally structured above the highest seasonable tide level, to prevent undesirable entrance of seawater into the saltworks area, while the bottom of the sluice gate was originally located at the level of paleo MSL, allowing the seawater flows into the saltwork area during high tides [5]. However, due to the lack of artifacts and organic material suitable for radiocarbon dating, the ages of their investigated medieval saltpans were mainly derived from historical archives [35]. Therefore, the current study first aims to reconstruct past environmental conditions based on analyses of the sedimentary sequence deposited in the saltwork area at Brbinj. The second objective is to detect evidence of man-made activities within this sequence and to determine the period of the salt-production activities at the location by dating the anthropogenic indicative layers. The third objective is to assess the potential of the used approach as a tool for reconstructing past sea levels. Consequently, the conducted study tested the potential of ancient, submerged saltpans as a new geoarchaeological approach for the inference of RSL data.

## 2 Regional setting: Geography, geology, archaeology and historical records

The indented eastern Adriatic coast (EAC) contains numerous submerged ancient saltwork constructions located usually in bays and coves facing north, protected from seasonal storms. Most of these ancient saltpans are well-preserved and dated to the medieval period by historical accounts [35–37].

The present study was conducted on the medieval saltpans in Lučina Cove next to the village of Brbinj, on the northeastern coast of Dugi Otok (meaning long island) in the Zadar archipelago (northern Dalmatia), EAC (Fig 1). The island, as most of the archipelago islands, became separated from the mainland 11,000 years ago due to the melting of late Pleistocene glaciers and the rising sea [38]. The island is ~44 km long and ~4 km wide, oriented from northwest to southeast, parallel to the Croatian coast and typical for EAC islands. Dugi Otok is a part of the Dinarides, belonging to the Adriatic NE unit or Dalmatian Karst, which is dominantly built of Mesozoic carbonates [39]. The island's geology includes limestones and dolomites, which display a typical karst morphology. Brbinj lies within the Albian-Aptian lithostratigraphic unit, which is part of the Sali-Božava anticline [40]. The anticline structure allowed the formation of a small karst polje filled with shallow red soil (*terra rossa*). According to the Croatian Soil Classification System and World Reference Base for Soil Resources, soils in the region are *terra rossa* soil and Chromic and Rhodic Cambisols [41].

Brbinj was first mentioned in the bull of Pope Celestin III in 1195. The document lists all properties owned by the Benedictine monastery of Saint Chrysogonus (Sveti Krševan) in Zadar, among them the church of Saint Damian in Brbinj, and its possessions [42, 43]. A subsequent document dated to 1196, describes the establishment of the saltwork site, when the

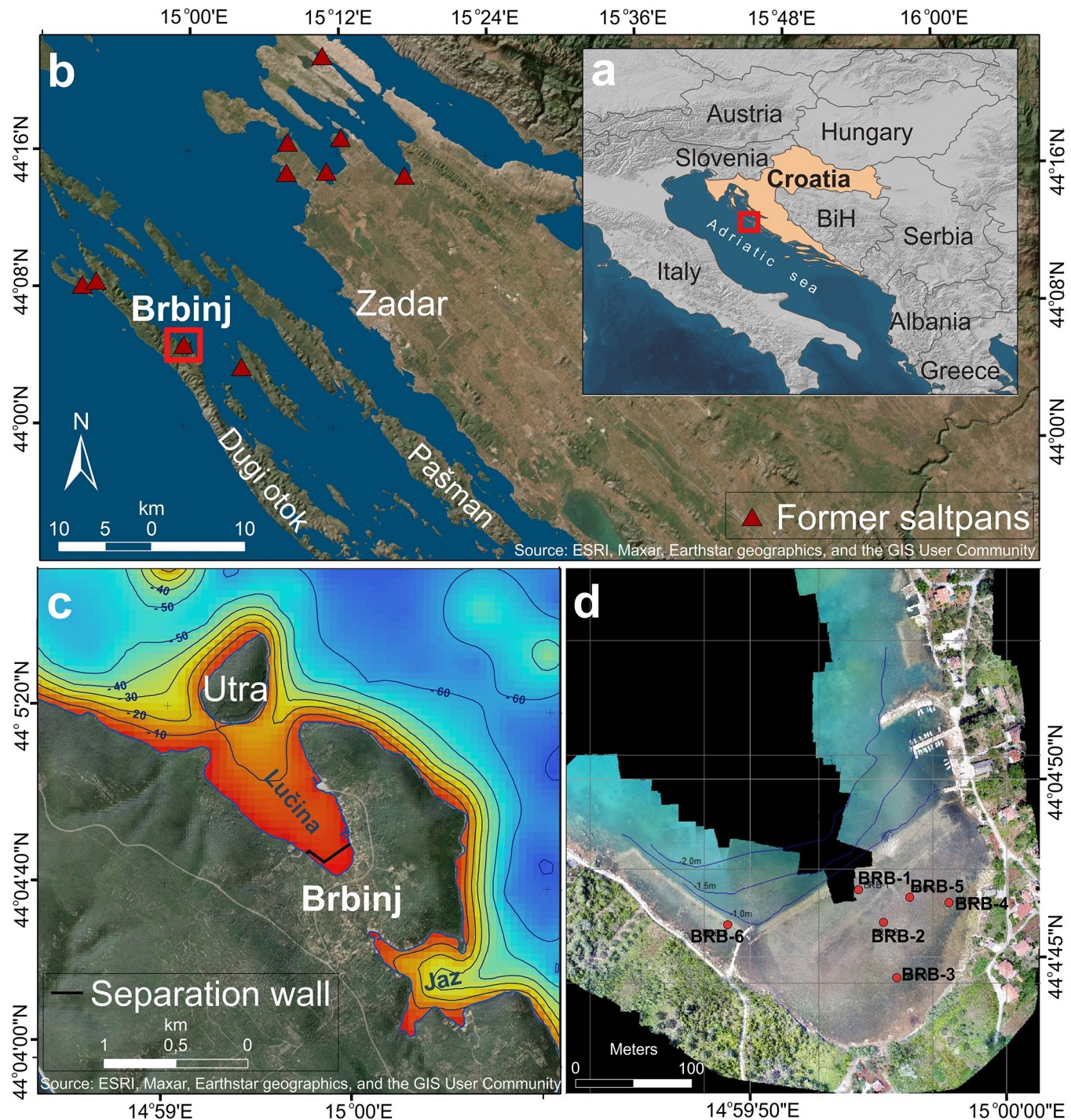

**Fig 1. Site map.** (a) Croatia and the Adriatic Sea. (b) Zadar archipelago and location of ancient saltpans. (c) Brbinj on Dugi Otok. (Source: ESRI, Maxar, Earthstar geographics, and the GIS User Community) (d) The saltpans area and the location of the cores (Source: digital surface model of the site generated by the authors, based on Bechor et al. [35]). The figures are similar but not identical to the original images and are therefore for illustrative purposes only.

abbot of the Saint Chrysogonus monastery rented one-third of the valley of Birbinio to the master Martino and his partner Matteo to build saltpans [36, 42, 44]. In this document, the monastery committed to finance the construction of the saltpans and stated their rights on salt

harvesting. In 1370, the abbot of Saint Chrysogonus abbey brought three salt workers from Trieste to rebuild and exploit the saltpans, which the monastery possessed in Brbinj [36, 42, 44]. Following the recognition of Venetian rule in 1409 and the monopolization of salt production and trade, most of the eastern Adriatic saltpans were abandoned or destroyed [36]. Most likely, also the saltpans of Brbinj were abandoned after this time.

The village of Brbinj (Fig 1C) stretches over two coves: Jaz to the south and Lučina to the north. The remains of the Brbinj saltpans are located in the shallow southern part of Lučina Cove, protected from the north by the small island of Utran/Utra. The saltpans are well preserved, and the remains of the separation wall, channels and at least six basins are still visible (Figs 1D and 2A). Previous archaeological exploration of the saltpan remains in Brbinj and an assessment of their relationship with local tide levels were conducted by Bechor et al. [35]. Their study used a combination of aerial photogrammetry, acoustic bathymetry scanning and an underwater archaeological survey of the site. The saltwork area in Brbinj has a U-shaped plan, with two narrow lateral branches connecting to the central part at an almost 90˚ angle (Fig 2A). Bechor et al. [35], assumed that seawater was channeled into the saltwork area from a sluice gate located on the western lateral branch, and was then distributed into the salt pools through a set of channels (Fig 2A). The basin on the western lateral branch was most likely the first evaporation basin and the following basins in the central part of the saltpans were most likely secondary evaporation basins. The southern two basins are probably the final evaporation pools from which highly concentrated brine was poured into the annexed crystallization pools (Fig 2A). The eastern lateral branch of the saltpan was destroyed by the construction of modern docks, preventing further interpretations.

Continuous and high-resolution elevation measurements of the separation wall and the sluice gate indicate the top of the separation wall is currently, on average, -75±9 cm below MSL and the bottom of the sluice gate is -99±9 cm (Fig 2B; [35]).

## 3 Materials and methods

Sedimentological, paleontological and geochemical analyses of sediment samples from three sedimentary cores were used to reconstruct the environmental conditions and footprint of manmade activities in the saltwork area. Following the identification of the saltwork layer in the sediment cores, OSL dating of prominent lithological boundaries was applied to determine the salt-production period. The depth and determined age of the anthropogenic layers were used to generate lower limiting points of past RSL.

### 3.1 Sediment cores

In total, seven sediment cores were collected in October 2018 from the saltpans area (Table 1 and Fig 1D) using an Eijkelkamp core sampler with a hand-auger set. The three cores BRB-1, BRB-2 and BRB-5 from three different pans in the saltwork area were examined in this study. The core BRB-1 with a length of 77 cm was retrieved from the outermost basin next to the separation wall where evaporation was initiated, whilst the cores BRB-2 and BRB-5 with lengths of 70 cm and 77 cm, respectively, were obtained from the inner pans (Figs 1D and 2A; Table 1). Two cores, BRB-1 and BRB-5 were scanned using a Philips Computerized Tomography (CT) scanner with an X-ray imaging unit in Carmel Medical Center, Haifa, Israel. The images of the scanned cores were generated using the MDC DiagNET application.

The three cores were sectioned lengthwise and photographed: cores BRB-1 and BRB-2 were sampled every 5 cm, whilst BRB-5 was sampled in 10 cm intervals for analysis of the particle-size distribution (PSD). PSD measurements, with a particle-size range of 0.04–2000 μm, were conducted with a Beckman Coulter LS 13 320 laser-diffraction particle-size analyzer. Each

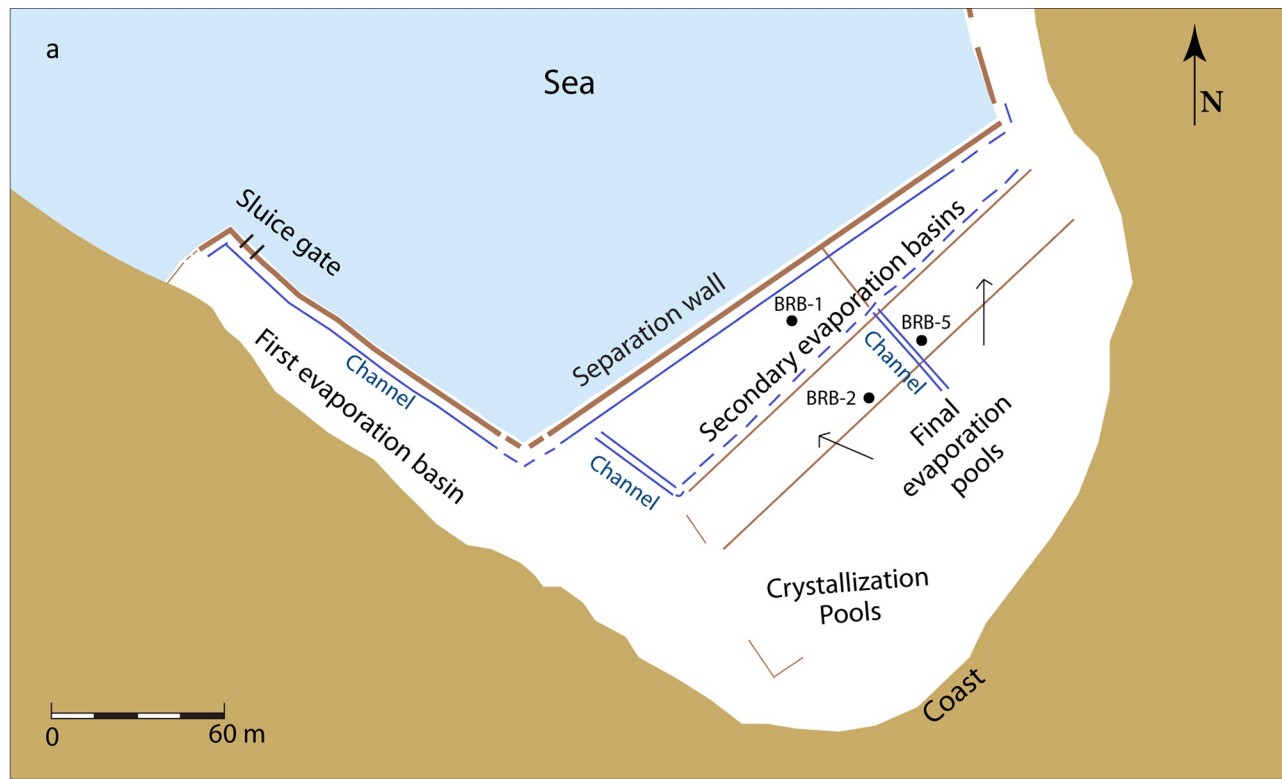

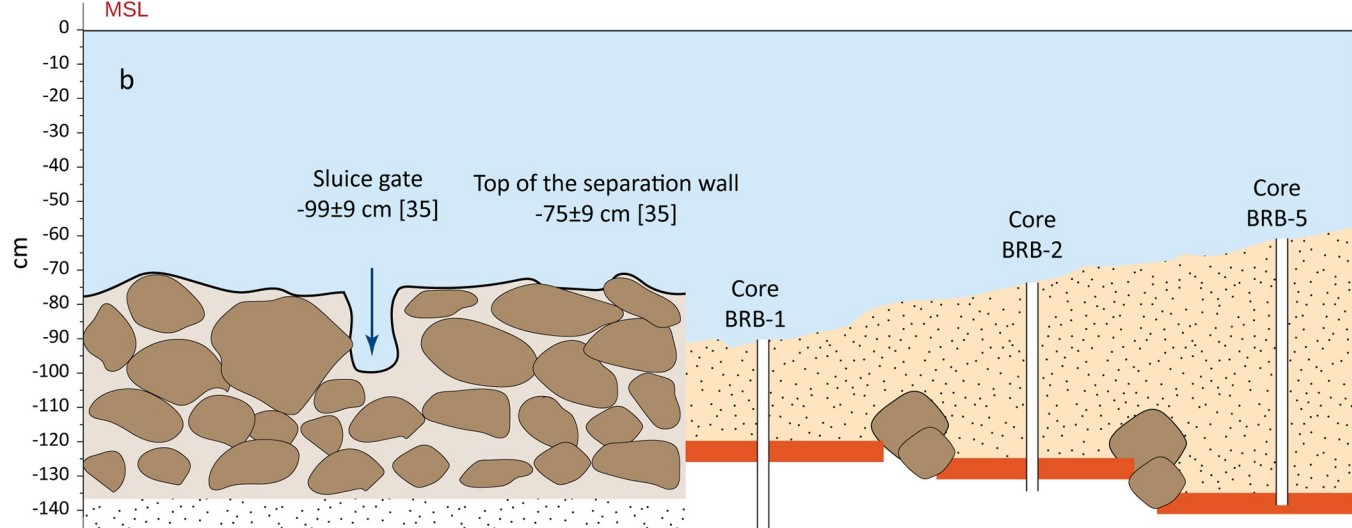

**Fig 2. Schematic sketch of the saltpan's structures and their relations to sea level.** (a) Schematic drawing of the surveyed saltworks site in Brbinj, the locations of the basins and the examined cores. (b) Present-day relations between MSL and manmade structures.

core was divided and logged into several sedimentary units according to their core depth, lithological classification and textural and structural characteristics, such as grain size, color, and the presence of calcareous macrofossils, and roots.

Uncertainty ranges of the unit's depths were evaluated following the protocols of Hijma et al. [45] and Vacchi et al. [7]. The calculated error of ±7 cm includes the sample-thickness error of the hand coring of ±5 cm and the obtained leveling error of ±5 cm using a high-

**Table 1. Main characteristics of the collected cores in Brbinj.**

| Core No. | HTRS coordinates | Length of retrieved core (cm) | Water depth, below current MSL (cm) | Core diameter (cm) | Comments |
|---|---|---|---|---|---|
| BRB-1 | N4883281 | 77 | -90 | 6 | Transparent PVC tube |
|  | E379735 |  |  |  |  |
| BRB-2 | N4883253 | 70 | -66 | 6 | Transparent PVC tube |
|  | E379757 |  |  |  |  |
| BRB-2 OSL | N4883253 | 90 | -66 | 5 | Non-transparent grey PVC tube |
|  | E379757 |  |  |  |  |
| BRB-3 | N4883204 | 48 | -33 | 6 | Transparent PVC tube |
|  | E379769 |  |  |  |  |
| BRB-3 OSL | N4883204 | 50 | -33 | 5 | Non-transparent grey PVC tube |
|  | E379769 |  |  |  |  |
| BRB-4 | N4883271 | 8 | -45 |  | Short core, sediments in a bag |
|  | E379814 |  |  |  |  |
| BRB-5 | N4883275 | 77 | -66 | 5 | Non-transparent grey PVC tube |
|  | E379780 |  |  |  |  |
| BRB-6 | N4883251 | 26 | -90 |  | Short core, sediments in a bag |
|  | E379622 |  |  |  |  |

accuracy GPS device. The angle error of the borehole was considered neglectable for the short cores.

## 3.2 Geochemical analysis

Total concentrations of major (Fe, Ca, Mg, Ti, Al, Na, K, P), minor (Mn) and trace (Mo, Cu, Pb, Zn, Cd, As, Sc, Ni, Co, As, Th, Sr, V, La, Cr, Ba, Zr, Ce, Nb, Y, Sc, Li and Rb) elements were determined from thirty-nine sediment samples of all three cores BRB-1, BRB-2 and BRB-5 using inductively coupled plasma emission/mass spectroscopy (ICP-AES/ICP-MS) in the ACME laboratory, Canada. The accuracy of the geochemical data is ±3% for major elements and 5 to 9% for minor and trace elements. The detection limit is 0.01% for major elements and in the range between 0.1 and 0.5 mg/kg for most of the trace elements. The element concentrations and their ratios to Ca in the reddish layers in the three cores from Brbinj (eight samples) were compared with geochemical properties of *terra rossa* soil from the Zadar area, including Dugi Otok (19 samples) published by Halamić et al. [46]. Comparability of new data and published geochemical data for soils was ensured by applying the same analytical method used by Halamić et al. [46].

## 3.3 Paleontological analysis

Thirty-one sediment samples of about 10 cm$^3$ from cores BRB-2 and BRB-5 were wet-sieved over 125 µm mesh and dried at 50°C for 24 hours. Samples were split into aliquots containing at least 50 foraminifera and ostracod valves representing the entire sample which were used for numerical analyses, following Heiri and Lotter [47] and Kemp et al. [48]. For samples containing less than the minimum number of foraminifera or ostracod valves, the entire sample was examined, and all specimens were counted (S1 and S2 Files). The reported specimens are stored for potential future analysis in the Croatian Geological Survey (HGI), Zagreb. Foraminifera and ostracods were mostly identified to species level, picked and counted under a binocular microscope to determine their numerical abundance (specimens per 10 cm$^3$ bulk sediment), the relative abundance (%) of the common species, the dominance and species

richness expressed as raw diversity. The foraminifera and ostracod assemblages were divided into groups according to the reported salinity-tolerance ranges of living species (S1 and S2 Files). Taxonomic identification of foraminifera followed Alve and Murray [49], Sgarrella and Moncharmont Zei [50], Jorissen [51], Avnaim-Katav et al. [52], Hayward et al. [53], Schönfeld et al. [54], and WoRMS [55]. Ecological groups follow Cimerman and Langer [56] and Hayward et al. [57, 58]. Taxonomic identification of ostracods is based on Bonaduce [59] and their assignment into ecological groups follows Salel et al. [60] and Mazzini et al. [61, 62]. Scanning electron microscope images of key foraminifera species were taken at the Department of Marine Geosciences, University of Haifa, and key ostracod species at the Institute of Earth Sciences, University of Iceland. In addition, mollusks were recorded in the cores and identified following the World Register of Marine Species (WoRMS) [55].

## 3.4 Optically Stimulated Luminescence (OSL) dating

The OSL method measures the elapsed time since mineral grains (primarily quartz) were last exposed to daylight. Four samples for OSL dating were taken from a non-transparent tube of BRB-2 (Table 1): BRB30 from 30–40 cm, BRB40 from 40–50 cm, BRB50 from 50–55 cm and BRB70 from 70–80 cm core depth. Samples were prepared and measured at the Luminescence Dating Laboratory in the Geological Survey of Israel (GSI), Jerusalem. Laboratory procedures were carried out under subdued orange light. Sediment samples for equivalent dose (De) in the size range of 74–125 µm were extracted using routine laboratory procedures [63]. After wet sieving to the selected grain size, carbonates were dissolved with 8% hydrochloric acid (HCl). The rinsed and dried samples were passed through a Frantz magnetic separator [64] to remove heavy minerals and most feldspars. Subsequently, the samples were soaked for 1 hour in 30% $H_2O_2$ at 50˚C to remove organic matter, followed by washing and drying in an oven. A 40-min rinse in 40% hydrofluoric acid dissolved the remaining feldspars and etched the quartz grains, followed by rinsing in 16% HCl overnight to remove fluorides which may have precipitated.

Sediments for dose-rate (d) evaluation were collected from the outer edges of the samples (about 1cm on each side). The sediments were dried and powdered, and the concentrations of U, Th and K were measured using ICP-MS (U and Th) and ICP-OES (K). Water contents were measured in two manners, either from the outer pieces removed from each sample or on an intact core sample from the same depth. For the former, water content was measured by weighing the sample immediately after extraction from the core liner and again after oven-drying at 50˚C. For the latter, the intact sample was dried, weighed, re-saturated with water and weighed again. In both cases, water contents are those for saturated samples. OSL measurements were carried out on a Risø DA-15 TL/OSL reader equipped with an integral 90Sr β source, with dose rates of 1.9 Gy/min. Stimulation was achieved with blue LEDs and detected through 7.5 mm U-340 filters. The single-aliquot regenerative-dose (SAR) protocol of Murray and Wintle [65] was used to determine the De values. Preheat and cut heat temperatures of 260˚C and 220˚C, respectively, selected after dose recovery tests showed that known doses can be recovered to within 95% certainty with such temperatures. Individual aliquots (19 for each sample) were measured, and the average De and errors were calculated using the central age model.

## 3.5 RSL lower limiting points evaluation

Marine limiting points are sediments deposited in open marine or lagoonal environments, indicating the limits of past sea levels, that do not meet the requirements of index points [7, 45]. The depth and age of the identified anthropogenic layer in ancient saltwork areas can produce marine lower limiting points of RSL.

# 4 Results

## 4.1 Core BRB-1: Sedimentological and faunal characteristics of sediment-core units

The core BRB-1 is divided into four units (Fig 3). The lower part of the core Unit IV, located at -152 to -159 cm below MSL (BMSL) is made of dark grey sand, partly including well-preserved shell fragments (*Cerithium repandum*) of up to 15 mm in size. Unit III (-134 to -152 cm BMSL) comprises reddish-grey silty sediment, with fragments of laminated material containing a few well-preserved gastropod shells of *Bittium reticulatum* and *Alvania* sp. of about 10 mm in length. Unit II (-108 to -134 cm BMSL) accommodates reddish sandy silt with a few small shell fragments such as *Odostomia nardoi*. The upper part of the core (Unit I, -90 to -108 cm BMSL) contains well-sorted grey sandy silt with shell fragments of *Tricolia pullus* and *Alvania lineata* smaller than 10 mm.

## 4.2 Core BRB-2: Sedimentological and faunal characteristics of sediment-core units

**4.2.1 Sediments.** Three units were observed along core BRB-2 (Fig 3). The lower part of the core (Unit III, -121 to -130 cm BMSL) contains reddish silty sediment with few unidentified shell fragments. Unit II (-110 to -121 cm BMSL) consists of reddish-brown sandy silt with a small number of shell fragments of *Cerithium lividulum* and *Steromphala albida*. The upper part of the core (Unit I, -66 to -110 cm BMSL) comprises dark brown sandy silt with relatively abundant and well-preserved gastropod shell fragments of mainly *Cerithium vulgatum*, in some cases up to 10 mm large. The upper 20 cm holds mainly plant debris, including roots.

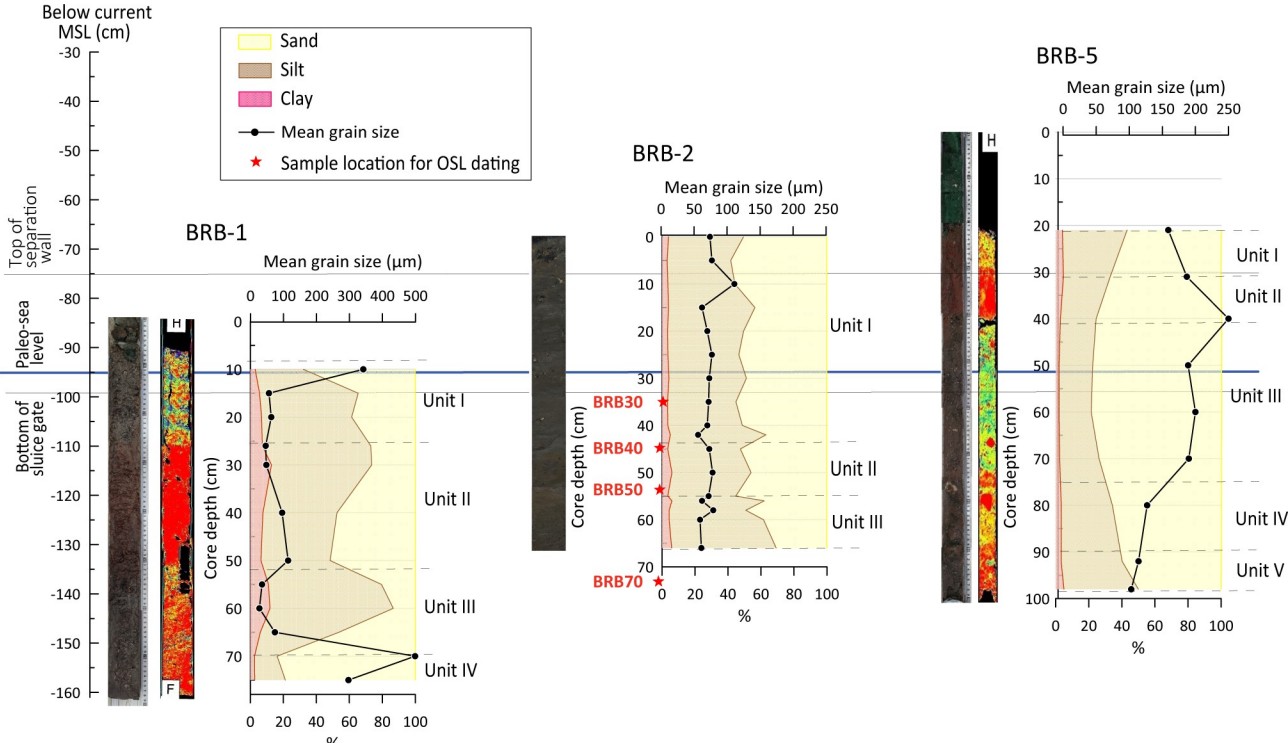

**Fig 3. The lithological units, CT, and grain size, of the cores BRB-1, BRB-2 and BRB-5.** The cores are aligned below the current MSL (BMSL) and presented in correlation with the top of the separation wall, the bottom of the sluice gate, and the derived paleo MSL.

**4.2.2 Foraminifera.** The benthic foraminiferal abundance ranges between 391 and 1422 specimens per 10 cm$^3$ (Fig 4). A total of 34 taxa were identified in core BRB-2 (S1 File). Species richness (per sample) varied between 8 and 16, with species belonging to the genus *Ammonia* as the most prevailing taxa representing 60–88% of the assemblage (Fig 5). The next common taxa are *Elphidium* spp. (5–20%), and *Ammobaculites* spp. and *Ammoscalaria runiana* (up to ~10% each). Higher abundances of brackish to marine species grouped under coastal taxa such as *Ammonia aberdoveyensis*, *Ammonia parkinsoniana*, and *Ammonia pawlowskii* characterize the sediments of units III and I. The sediments of Unit II are characterized by a significant increase of the euryhaline species *Ammonia veneta* [53] dominating the foraminiferal assemblage (66–76%) parallel to the sharp reduction in the abundance of coastal taxa (Fig 4).

**4.2.3 Ostracods.** The ostracod abundance varies between 2 and 232 valves per 10 cm$^3$ sediment. Sediments of units I and III contain few valves, while Unit II contains a relatively large number of valves (Fig 4; S2 File). Species richness varies between 2 to 11 per sample, with the euryhaline *Cyprideis torosa* as the most abundant taxon (24–90%). The next common taxa are *Xestoleberis communis* (up to ~50%), *Aurila* spp. (up to ~25%) and *Loxoconcha* spp. (Fig 6). The abundance of oligohaline to euhaline taxa grouped as coastal species including *Xestoleberis communis*, *Aurila* spp., *Loxoconcha* spp. and *Callistocythere* cf. *adriatica* fluctuates in Unit II and is more or less similar to the abundance of the euryhaline species (Fig 4).

## 4.3 Core BRB-5: Sedimentological and faunal characteristics of sediment-core units

**4.3.1 Sediments.** Core BRB-5 is divided into five units (Fig 3). The lower part of the core (Unit V, -133 to -140 cm BMSL) contains laminated silty material of alternating more reddish and darker colors, including a few dark stains. Unit IV (-119 to -133 cm BMSL) is reddish-

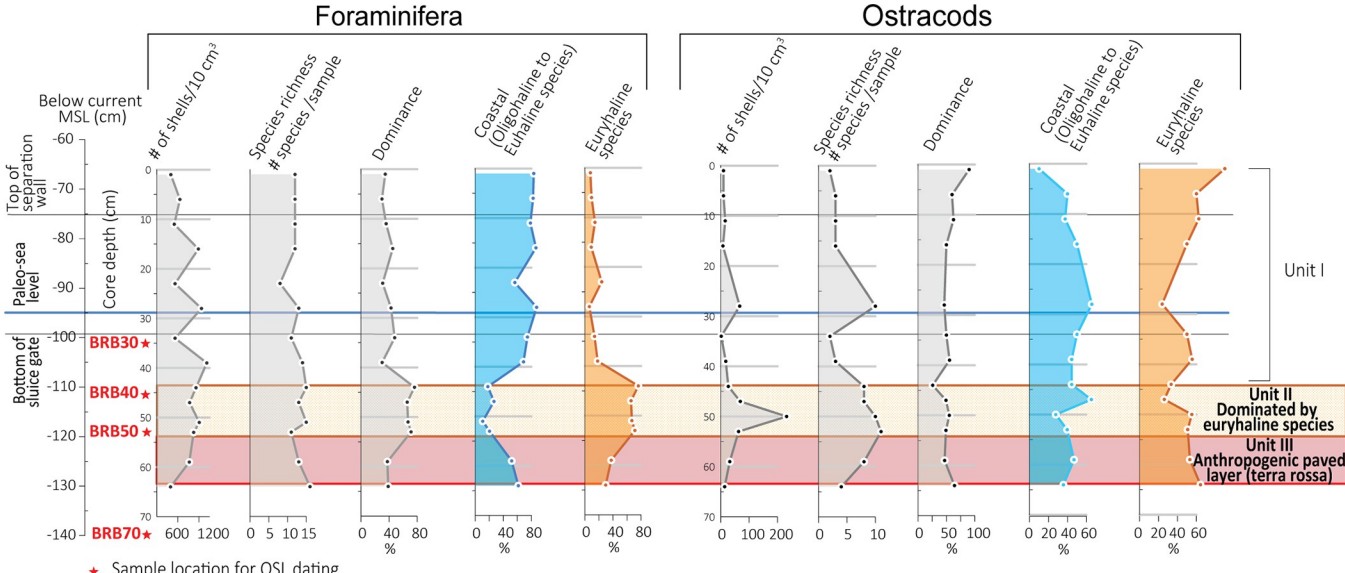

**Fig 4. Foraminiferal and ostracod characteristics of core BRB-2 vs. current MSL.** Abundance (number of shells/10 cm$^3$), species richness (number of species/sample), and dominance (%). Relative foraminiferal abundances of coastal species (*Ammobaculites* sp., *Ammoscalaria runiana*, *Ammonia aberdoveyensis*, *Ammonia parkinsoniana*, *Ammonia pawlowskii*, *Cribroelphidium poeyanum*, *Cribroelphidium williamsoni*, *Elphidium aculeatum*, *Elphidium advenum*, *Elphidium crispum*, and *Elphidium limbatum*) vs. euryhaline species (*Ammonia veneta*) based on Alve and Murray [49], Avnaim-Katav et al. [52], Hayward et al. [53] and Schönfeld et al. [54]. Relative ostracod abundances of coastal species (*Xestoleberis communis*, *Aurila prasina*, *Aurila woodwardi*, *Leptocythere lagunae*, *Loxoconcha* cf. *elliptica* and *Loxoconcha ovulata*) vs. euryhaline species (*Cyprideis torosa* f. *littoralis*) based on Bonaduce [59] and Mazzini et al. [61].

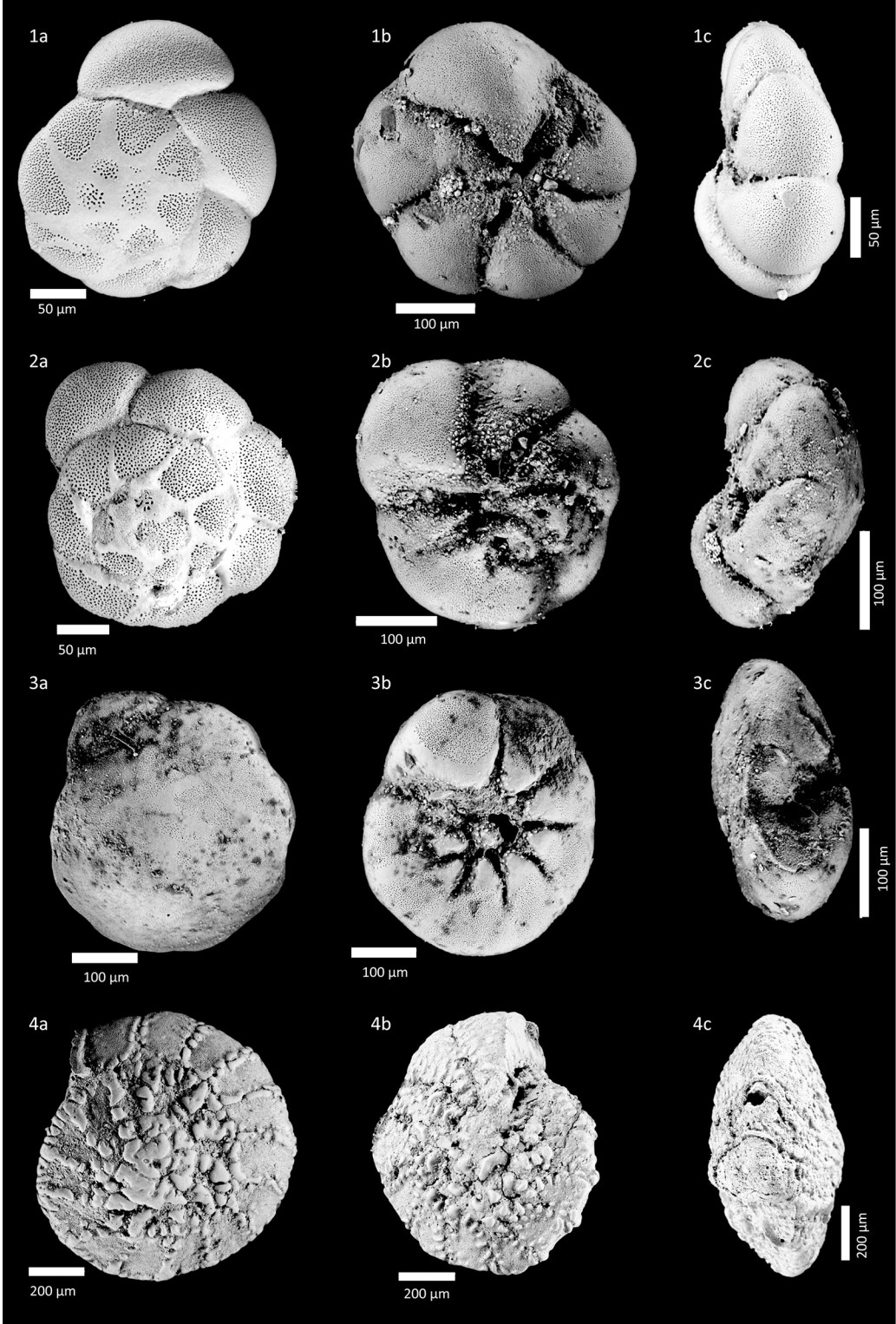

**Fig 5. Foraminifera from core sediments collected in Brbinj, Dugi Otok, Dalmatian coast, Croatia.** (1a–c) *Ammonia aberdoveyensis* Haynes 1973, (a) Spiral side view, (b) Umbilical side view, (c) Profile view; (2a–c) *Ammonia veneta* (Schultze 1854), (a) Spiral side view, (b) Umbilical side view, (c) Profile view; (3a–c) *Ammonia parkinsoniana* (d'Orbigny 1839), (a) Spiral side view, (b) Umbilical side view, (c) Profile view; (4a–c) *Ammonia pawlowskii* Hayward and Holzmann 2019, (a) Spiral side view, (b) Umbilical side view, (c) Profile view. Note: we used the sutures characteristics and the test

profile as morphological features for distinguishing *A. aberdoveyensis* from the most similar non-ornamented species *A. veneta* [53, 54]. Sutures on the domed spiral side of *A. veneta* are flushed, raised and covered by calcite ridges ornament, whereas *A. aberdoveyensis* is characterized by a weakly convex to flat spiral side with its smoother sutures.

grey silt with rounded pebbles in a size of 30–35 mm. Unit III (-85 to -119 cm BMSL) consists of dark silty sediment with abundant well-preserved shell fragments mainly belonging to *Cerithium lividulum* and *Odostomia nardoi*. Unit II (-75 to -85 cm BMSL) represents homogeneous reddish sandy silt without macro-fauna remains. The upper part of the core (Unit I, -66 to -75 cm BMSL) contains poorly sorted reddish sandy silt, granules (up to 3 mm), and up to 12-mm long shell fragments of *Cerithium vulgatum*.

**4.3.2 Foraminifera.** The foraminiferal abundance varies in a relatively large range between 34 and 1228 specimens per 10 cm$^3$ (Fig 7). A total of 55 benthic foraminiferal taxa were identified and species richness varies between 2 and 21 species per sample (Fig 7; S1 File). The predominating taxa are *Ammonia* species (40–100%). Other abundant taxa are *Quinqueloculina* spp. (up to ~27%) and *Elphidium* spp. (up to ~20%). The sediments of Unit V contain mostly scarce and poorly preserved brackish to marine foraminifera. Relatively low abundance ranging between 200 and 416 specimens per 10 cm$^3$ characterizes the overlying sediments of Unit IV, which are dominated by *A. veneta*. The upper sedimentary units III to I show maximum foraminiferal abundances of coastal species, where the most dominant ones are *A. aberdoveyensis*, *A. parkinsoniana* and *A. pawlowskii* (Fig 7).

**4.3.3 Ostracods.** The ostracod abundance varies between 2 and 760 valves per 10 cm$^3$ sediment (Fig 7). In total, 21 species were identified with a species richness between 1 and 11 per

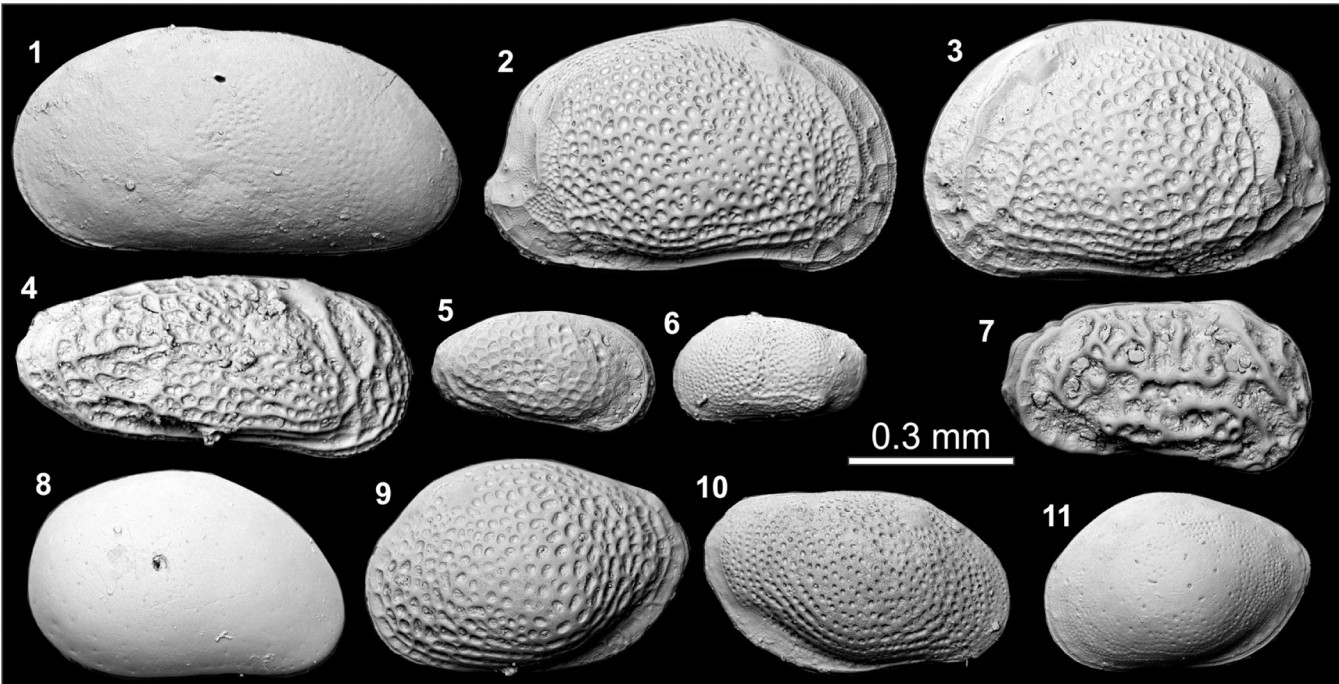

**Fig 6. Ostracod valves from core sediments collected in Brbinj, Dugi Otok, Dalmatian coast, Croatia.** (1) *Cyprideis torosa* f. *littoralis* (Jones 1850) left valve (LV); (2–3) *Aurila woodwardi* (Brady 1868), (2) right valve (RV), (3) LV; (4) *Leptocythere lagunae* Hartmann 1958 RV; (5) *Leptocythere* cf. *bacescoi* (Rome 1942) RV; (6) *Leptocythere* cf. *bituberculata* Bonaduce, Ciampo and Masoli 1976 LV; (7) *Callistocythere* cf. *adriatica* Masoli 1968 RV; (8) *Xestoleberis communis* Müller 1894 RV; (9) *Loxoconcha rhomboidea* (Fischer 1855) LV; (10) *Loxoconcha* cf. *ovulata* (Costa 1853) RV; (11) *Loxoconcha* cf. *elliptica* Brady 1868 juvenile, LV.

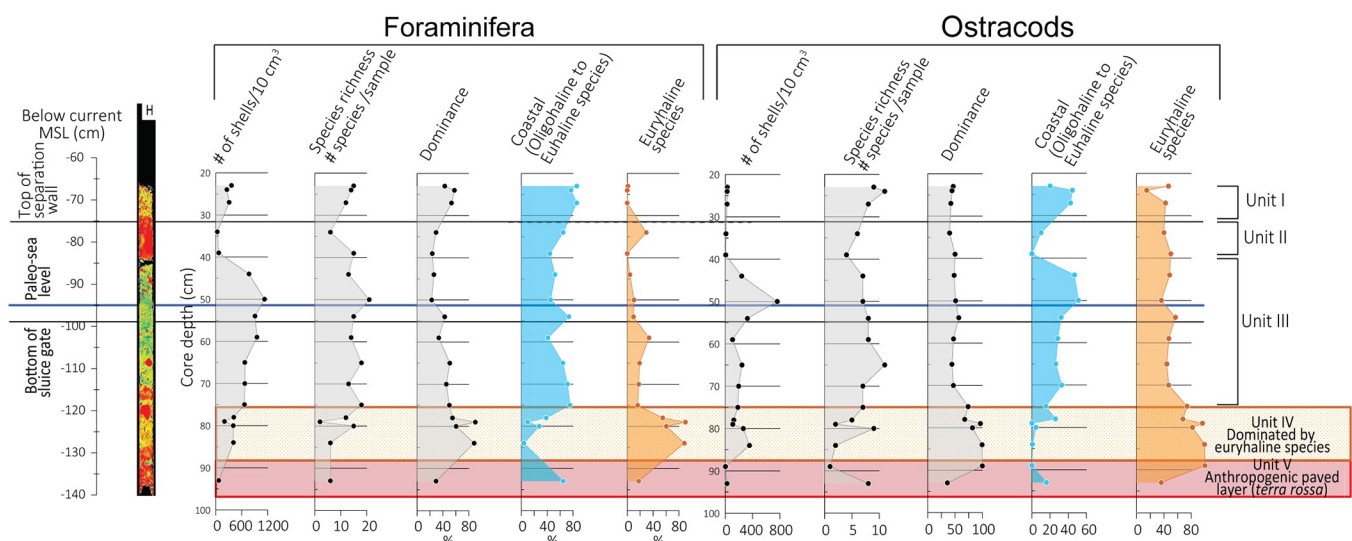

**Fig 7. Foraminiferal and ostracod characteristics of core BRB-5 vs. current MSL.** Subdivision into units used in the text are denoted and explained in the caption of Fig 4.

sample. *Cyprideis torosa* is the most prevailing taxon occupying 11–100% of the assemblage (S2 File). The next common taxa are *Aurila* spp. (up to ~50%), and *Xestoleberis communis* (up to ~30%). Sediment Unit V contains scarce and poorly preserved specimens dominated by typical coastal taxa. Sediment Unit IV is predominated by *Cyprideis torosa* (74–99%) and is characterized by an increase in the abundance of coastal ostracods. The upper sedimentary units III to I include abundant coastal specimens such as *Xestoleberis communis* and *Aurila prasina* (Fig 7).

## 4.4 Geochemical characteristics

Geochemical properties of the thirty-nine sediment samples from cores BRB-1, BRB-2 and BRB-5 indicate a decreasing trend in Ca concentrations with core depth in correlation to an opposite trend of increasing Al and Fe concentrations (S3 File). These trends of Al and Ca concentrations with core depth correspond to increases in silt contents and decreases in carbonate contents, respectively. Concentrations of elements of terrigenous origin such as Fe, K, Ti and Al, normalized to Ca, are high and indicate the prevailing deposition of detrital sediments (Fig 8). Peaks in Mg/Ca, K/Ca, Fe/Ca, Ti/Ca and Al/Ca ratios occur in the reddish unit II of core BRB-1 at -108 to -134 cm BMSL, in the reddish unit III of core BRB-2 at -121 to -130 cm BMSL, and in the reddish unit V of core BRB-5 at -133 to -140 cm BMSL (Fig 8). The positions of these peaks in the three cores result in a shallow slope from BRB-1 in the outermost basin of the saltpan structure close to the separation wall down to BRB-2 and further down to BRB-5 in the inner basins (Figs 1D, 2A and 8).

Comparison of the geochemical properties of the three reddish layers in the examined cores from Brbinj with geochemical properties of the *terra rossa* soil from the Zadar area exhibits almost similar median values for the concentrations of the trace elements in the samples from Brbinj and those from the Zadar area (Fig 9).

## 4.5 Dating

OSL-dating results of the indicative units in core BRB-2 are summarized in Table 2. The sample BRB70 from below the reddish Unit III (-141 cm BMSL) is dated to 240±100 CE. The

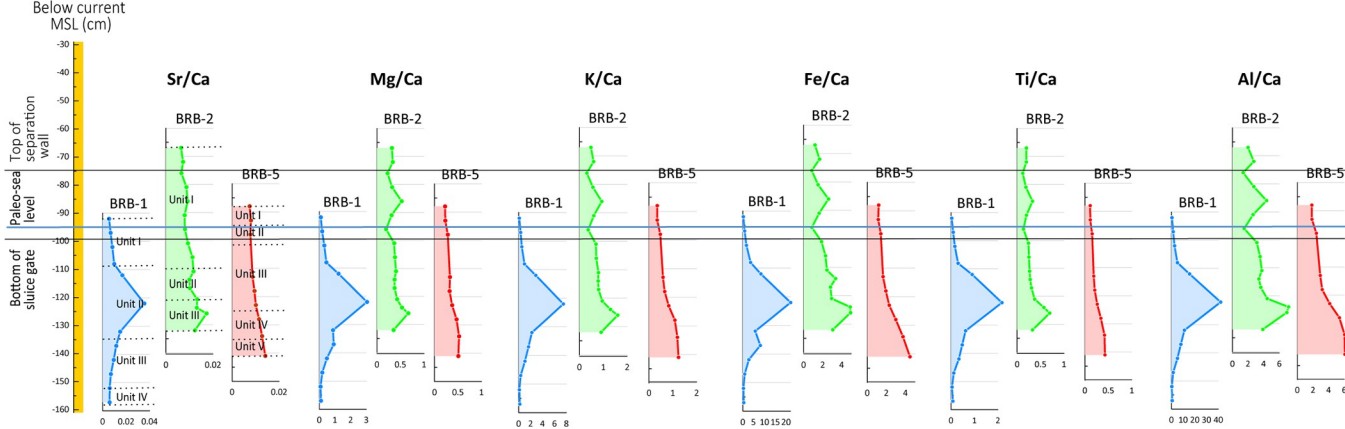

**Fig 8. Variations in terrigenous elements ratios to Ca in cores BRB-1, BRB-2 and BRB-5.** Elevations are relative to the current MSL.

sample BRB50 from the lower part of Unit II immediately above the reddish Unit III (-119 cm BMSL) corresponds to the bottom of the euryhaline layer, is dated to 1040±50 CE. The sample BRB40 from the upper part of Unit II (-111 cm BMSL), corresponding to the top of the euryhaline layer, is dated to 1390±30 CE. The sample BRB30 from the lower part of Unit I (-101 cm BMSL) and a position corresponding to the height of the sluice gate is dated to 1540±20 CE (Figs 3 and 4).

## 5 Discussion

Following the study objectives to identify the salt-production units in the drilled cores, two main strategies were regarded as most appropriate: a) Geochemical (element concentration) data as tracers for identification of a terrestrial unit regarded as *terra rossa* pavement brought artificially from the near surroundings and b) Paleontological (Foraminifera and Ostracoda assemblage) data including species typical of euryhaline environments, indicating the salt-production unit. Following the detection of the saltwork unit, the unit was dated and used as a lower limiting point of RSL (Fig 10).

### 5.1 Paleoenvironmental interoperations of the site

The units that most likely represent sediments accumulated at a time when the saltworks location was a natural bay or potentially a lagoon are Units III and IV in core BRB-1. The units are characterized by dark gray sediment with a mean grain size of up to 500 μm (Fig 3), containing large and preserved macrofauna typical for a brackish environment. OSL dating of a sample below Unit III in core BRB-2, which correlates with the same depth as Unit III in core BRB-1 (Fig 3), revealed an age of 240±100 CE for the pre-saltwork phase.

The saltwork phase is represented by the reddish layers of Unit II in core BRB-1 and the base of the two inner cores: Unit III in core BRB-2 and Unit V in core BRB-5. The layers were identified to originate predominantly from terrigenous sediments based on their geochemical composition comparable to *terra rossa* soils, reddish color, and fine-grained characteristics. These layers of ~10 cm thickness are located at slightly different depths, ranging from -120 ±7 cm BMSL in the pool nearby the separation wall (Unit II in BRB-1), to -125 ±7 cm BMSL in the inner south-western pool (Unit III in BRB-2) and to -135 ±7 cm BMSL in the inner south-eastern pan (Unit V in BRB-5). The determined shallow gradient is inclined from the outer pan into the inner pans (Figs 8 and 10), consistent with a natural flow of the brine from low to

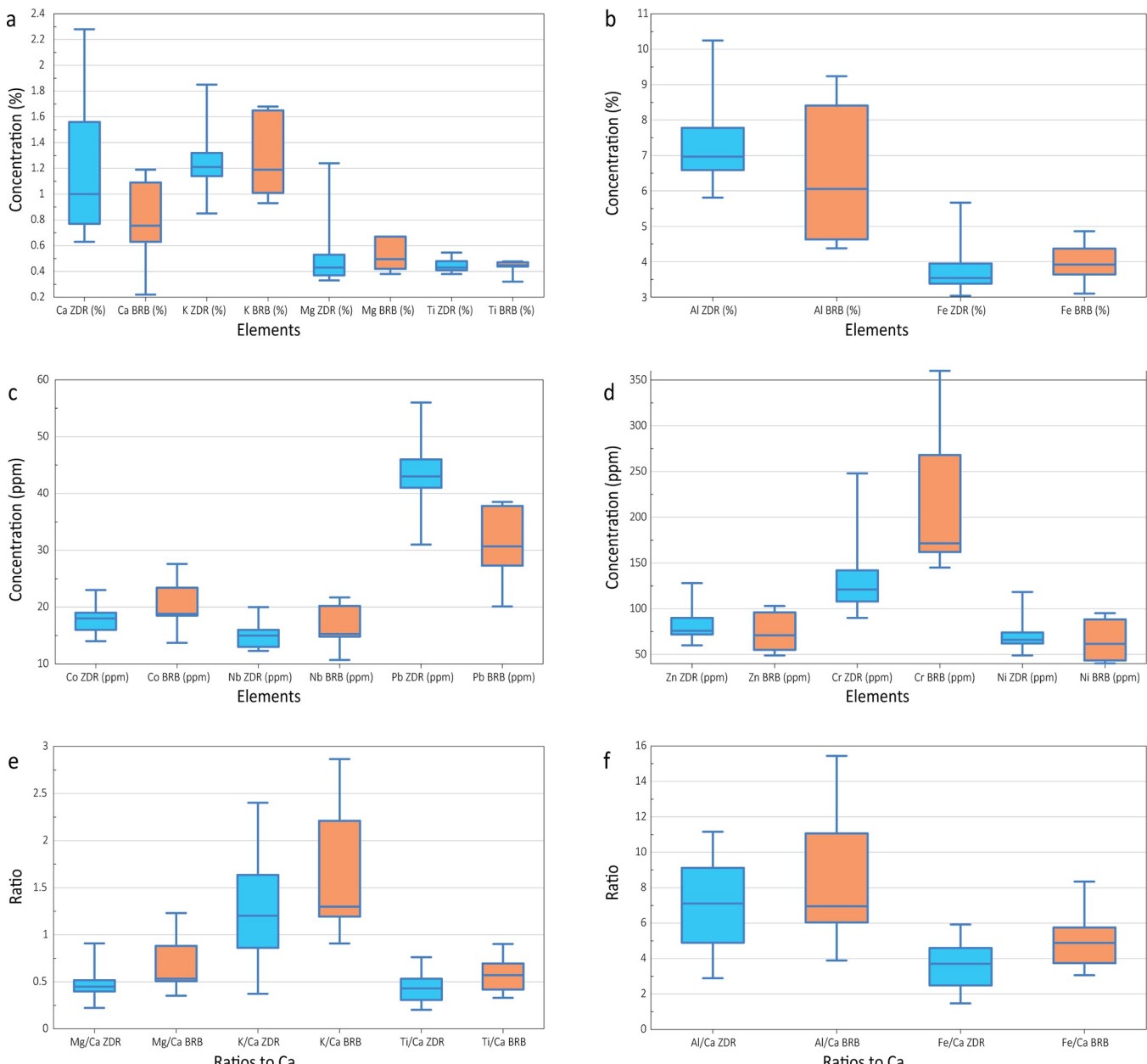

**Fig 9. Comparison of elements concentrations and their ratios to Ca, between current study samples from the reddish layers in Brbinj (BRB) to *terra rossa* soil from Zadar area (ZDR), following Halamić et al. [46].** (a) Concentrations of Ca, K, Mg and Ti. (b) Concentrations of Al and Fe. (c) Concentrations of Co, Nb and Pb. (d) Concentrations of Zn, Cr and Ni. (e) Ratios of Mg/Ca, K/Ca, and Ti/Ca. (f) Ratios of Al/Ca and Fe/Ca.

high concentration pools. The identified uniform reddish layers in the three cores likely represent a man-made paved layer as part of the saltpan structure. *Terra rossa* is a reddish silty-clayey soil material widespread in the Mediterranean region which covers limestone and dolomite as a discontinuous layer ranging in thickness from a few centimeters to several meters [66]. Thick accumulations of *terra rossa* are situated in karst depressions in the form of pedo-sedimentary complexes [66]. Comparison between the geochemical properties of the reddish units in the three examined cores, and *terra rossa* soil from the Zadar area including the karst polje in the hinterland of Brbinj [46] verify that the sediments of the reddish layer in the Brbinj

**Table 2. Summary of OSL dating.**

| Sample | Core depth (cm) | Average position below MSL (cm) | Moisture contents (%) | K (%) | U (ppm) | Th (ppm) | Dose rates (Gy/ka) | N | De (Gy) | Age (Years before 2020) | Calendar Age (CE) |
|--------|-----------------|-------------------------------|----------------------|-------|---------|----------|--------------------|----|---------|-------------------------|-------------------|
| BRB30 | 30–40 | -101 | 72±7 | 1.10 | 15.4 | 17.0 | 3.51±0.15 | 18/19 | 1.67 ±0.04 | 480±20 | 1540±20 |
| BRB40 | 40–50 | -111 | 63±6 | 1.13 | 16.2 | 17.5 | 3.85±0.15 | 19/19 | 2.44 ±0.05 | 630±30 | 1390±30 |
| BRB50 | 50–55 | -119 | 55±5 | 1.21 | 11.2 | 16.3 | 3.30±0.12 | 18/19 | 3.21 ±0.10 | 980±50 | 1040±50 |
| BRB70 | 70–80 | -141 | 16.5±3 | 1.00 | 5.1 | 15.9 | 3.01±0.14 | 18/19 | 5.37 ±0.14 | 1780±100 | 240±100 |

cores mostly originated from local *terra rossa* materials (Fig 9). The median values for the trace-element concentrations of samples from Brbinj are almost similar to those for *terra rossa* from the Zadar area, apart from the concentration of Pb which is significantly higher in the Zadar area. However, the higher concentrations of Pb in *terra rossa* result from modern pollution [29], whilst lower concentrations are observed in the *terra rossa* materials buried more than 800 years ago at the saltwork area of Brbinj. Previous studies already suggested that ancient saltpans were underlined by a layer of clayey silts which consist of terrigenous sediments of *terra rossa* [29, 67, 68]. However, these studies did not suggest that the *terra rossa* deposits at the base of saltpan sequences potentially resulted from anthropogenic activities during the construction of the saltpans. An example of man-made pavements at the base of saltpans exists on the bottom of the crystallization pools in Sečovlje (Piran) salina located in the Slovenian Istrian Peninsula, northern Adriatic Sea [5]. Here, fresh marine mud was used to generate the "petola", a permanent microbial mat which is cultivated at the bottom of the crystallization basins.

A barrier such as the separation wall enables the control of the supply of seawater to more evaporated water, finally turning to hypersaline brine in the artificially created setting of the saltworks [4, 5]. As a result, different physico-chemical conditions and related salinity levels prevailed in the pools supporting specific microfauna assemblages whose remains represent powerful tools for the reconstruction of past environmental conditions [27]. The sedimentary unit overlying the paved layer in the inner pools (Unit II in BRB-2 and Unit IV in BRB-5) of the saltworks area includes abundant shells of the euryhaline foraminifera *Ammonia veneta* [53], which dominate the assemblage (Figs 4 and 7). Correspondingly, the predominant ostracod species observed in this layer is *Cyprideis torosa* (Figs 4 and 7), a species that tolerates

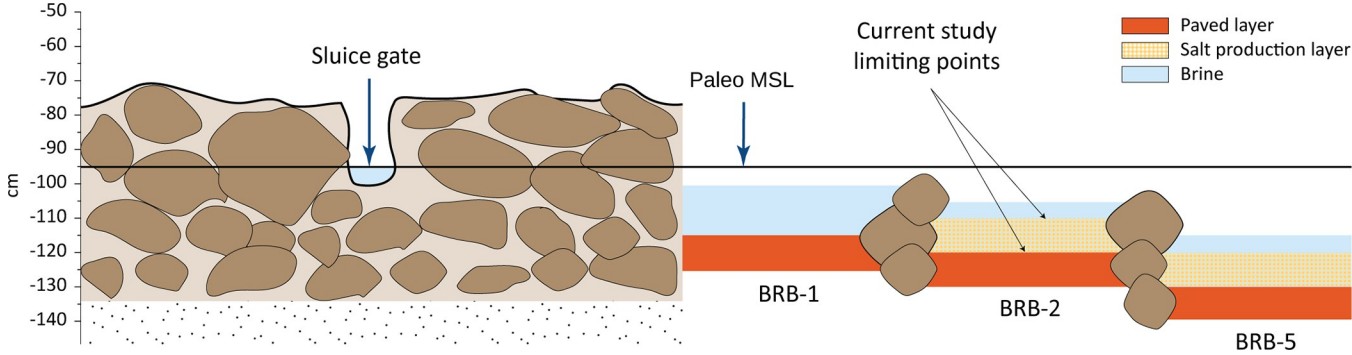

**Fig 10. Schematic reconstruction of the saltpans in Brbinj during its operational period.** Past sea-level, based on elevations of the sluice gate bottom and the separation wall top [35]. The paved, euryhaline layers and the RSL lower limiting points, were identified in the current study.

euryhaline conditions [28, 61]. The presence of prevailing euryhaline taxa in a layer above the paved unit strengthens the interpretation of this unit as the saltworks unit.

A high abundance of *Ammonia tepida* Cushman in hypersaline environments was observed in the Dead Sea pools by Almogi-Labin et al. [69], and in the northern Aegean Sea at the Camalti saltpan (still operating) by Meriç et al. [27], and in the saltpans, hypersaline lagoons and salt lakes of the Gulf of Saros by Bassler-Veit et al. [70]. These observations were made prior to the new definitions of the *Ammonia* clade by Hayward et al. [53]. *Ammonia veneta* observed in the current study, belongs to the *tepida* morpho group, which includes mostly taxa that can tolerate hypersaline conditions [53]. The highest abundances of *Cyprideis torosa*, a species adapted to hypersaline waters, have been observed also by Meriç et al. [27] and Barut et al. [28] in the saltpans of the northeastern Aegean Sea, and by Mazzini et al. [61] in the Italian Peninsula.

The boundary between the *terra rossa* layer and the overlaying euryhaline layer represents the time when the salt production was initiated, dated by OSL to 1040±50 CE, which is almost a century earlier than the period inferred from historical documents which provide discontinuous, sporadic information on the region. The sediments at the top of the euryhaline layer most likely represent the end of the salt-production period, dated by OSL to 1390±30 CE, which correlates with the historical archives within the error limits [36, 44, 71].

The silty-sandy units overlaying the euryhaline layers (Unit I in core BRB-2 and Units I-III in core BRB-5), were accumulated during the phase when the saltwork area was abandoned and covered by the rising sea. These units are rich with a large variety of marine species such as *A. aberdoveyensis*, *A. parkinsoniana*, *A. pawlowskii*, *Xestoleberis communis* and *Aurila prasine*. OSL dating of a sample from the bottom of Unit I in core BRB-2 (- 100 cm BMSL) correlating with Unit III in core BRB-5 (Fig 3) yielded an age of 1540±20 CE.

## 5.2 RSL inferences

Last millennium sea-level data obtained in previous studies at sites located in proximity to the current study are mostly lower in comparison to reconstructed RSL models (Fig 11). For example, Marriner et al. [10] identified RSL index points based on sedimentary proxy data ranging from -50 to -90 cm for the period between the 11th and 13th centuries. In contrast, Faivre et al. [33] suggested even lower RSL using biogenic littoral rims built by the coralline rhodophyte *Lithophyllum*, but the authors attributed the relatively low levels for this period to probable contamination of the radiocarbon-dating materials and resulting biases of age data. Bechor et al. [35] investigated medieval saltpans in central Dalmatia as archaeological sea-level indicators and concluded that the RSL was -50±12 cm at 840±250 CE in Vrgada Island, -92±8 cm at 1350±50 CE in Lavsa Island and -95±9 cm at 1300±106 CE at Brbinj (Fig 11). The ages of RSL at these three sites relied primarily on historical documents. In contrast, the current study provides direct age evidence for the use of the saltworks at Brbinj by the identification of the anthropogenic layer in the sedimentary sequence and the determination of its age to 1215±175 CE (the mean value between the time when the salt production was initiated, dated by OSL to 1040±50 CE and the date when the salt-production ceased, dated by OSL to 1390±30 CE) (Fig 11).

The reconstructed RSL model predictions are based on the Australian National University (ANU) and ICE-6G C(VM5a) models [72, 73], respectively (Fig 11). The two applied models differ since they rely on different continental ice sheet and mantle viscosity profiles [74]. Almost all proxy-based RSL reconstructions in the study area are lower than those based on modeling approaches. Shaw et al. [34] suggested that disagreements between proxy-based RSL and model-based reconstructions can be attributed to the tectonic subsidence of the EAC. However, vertical crustal movement determined by collocated continuous GPS stations along

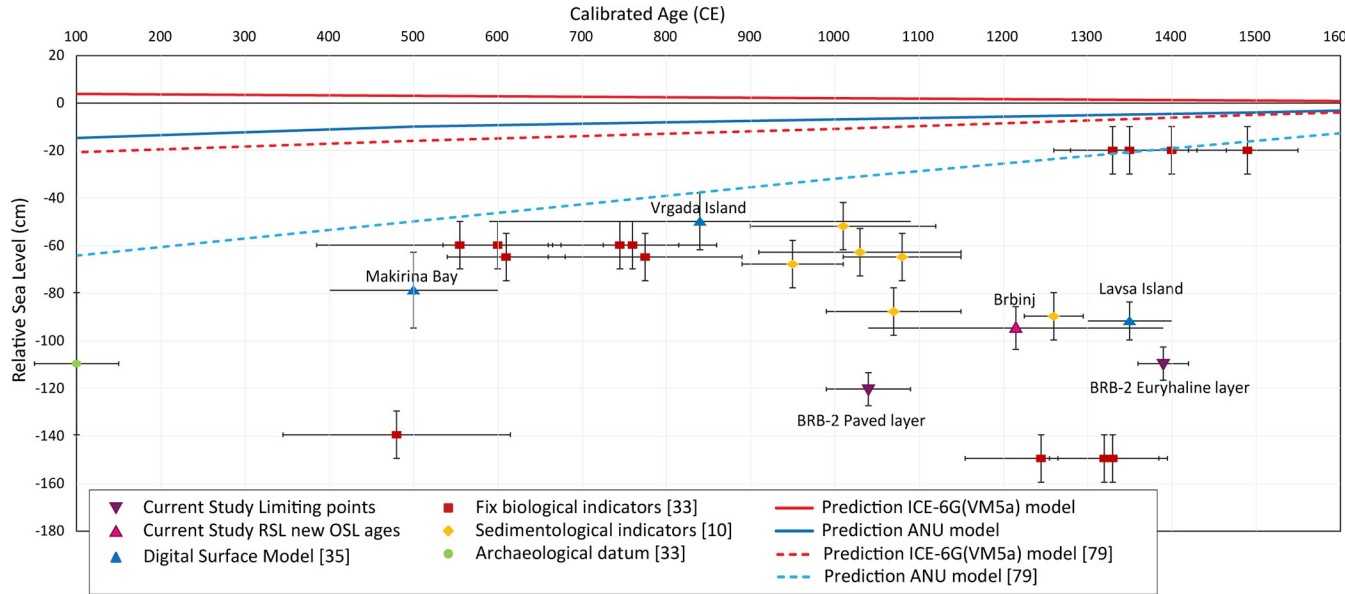

**Fig 11. Reconstruction of RSL in central Dalmatia, during the last two millennia.** Our newly suggested limiting points of RSL at Brbinj and the determined OSL age, in comparison to previously presented reconstructions and inferences of glacio-isostatic prediction models [35, 79].

the EAC indicate that the northern part is characterized by low subsidence rates during the last few thousand years, in contrast to subsidence of -1.7 mm/a in the southern Adriatic coast [75]. The central part of the Dalmatian coast seems to be more stable during long periods [7]. Similarly, Pace et al. [76] suggest that present-day GPS measurements point to lower kinematic movement of the northern Adria plate, and 5mm/a movement northward in the southern Adria plate, with no modelled data for the mid-Adriatic region. In addition, calculated GIA rates based on the ANU model vary between -0.17 to -0.22 mm/a [35]. Therefore, all the RSL indications presented in Fig 11 can be explained either by significant vertical subsidence due to earthquakes after the abandonment of the saltwork activities or by fluctuating sea level in time slices that cannot be predicted by the GIA models as suggested by Bechor et al. [35]. The European Archive of Historical Earthquakes [77, 78] recorded two strong earthquakes (Mw ≈ 6.0) following the saltwork time period, one in April 1418 and another one in July 1717. However, the underwater survey of Bechor et al. [35] did not identify signs of destruction or vertical movement indicative of significant subsidence in the saltpan constructions.

The presented study not only suggests a new method for obtaining past sea level, it also established two new marine lower limiting points: -125±7 cm at 1040±50 CE and -110±7 cm at 1390±30 CE (Fig 11), which constrain the RSL in Brbinj. It also revealed that the bottoms of the saltpan pools were paved with hinterland *terra rossa* materials. The new method obtains RSL limiting points in saltpans that were not excavated or studied before, based on coring in the pan's remains behind the separation wall, identifying the anthropogenic sedimentary units and determining their age. The well-recognized anthropogenic units can be used to determine RSL marine limiting points; this method provides a new approach that can be applied in many saltpan locations around the Mediterranean and elsewhere.

## 6 Conclusions

The study in Brbinj combines for the first time sedimentological, paleontological and geo-chemical analyses of sediment cores from submerged medieval saltpans, aiming to detect

various environmental conditions as evidence for saltwork activities. The study determined the depths and the ages of the salt-production unit and reconstructed limiting points of paleo sea level. As a novelty, the study identified terrigenous materials of *terra rossa* soil in the examined saltpan basins, which most likely represent an artificially created paved layer. The sediments which overlay the *terra rossa* unit are rich in faunal remains dominated by euryhaline species such as *Ammonia veneta* and *Cyprideis torosa*. These deposits represent the hypersaline environment that existed during the saltwork activity. Following the abandonment of the saltpans, the area was flooded by the rising sea, as indicated by the deposition of an upper silty-sandy unit including a variety of typical marine organism remains. The transitions between the paved and euryhaline layers, and the euryhaline and marine layers, were dated using OSL which unveiled the salt production period. The lower marine limiting points of RSL established are -120 ±7 cm, dated to 1040±50 CE and -110 ±7 cm, dated to 1390±30 CE. Both limiting points constrain the RSL at Brbinj. Based on the applied multi-proxy approach, this study was able to recognize and date the anthropogenic unit related to salt production in the sedimentary sequence of the cores. The saltpan floor and the organism remains indicating hypersaline conditions during the deposition of the overlying sedimentary unit were used here as a new tool to infer RSL change in medieval times. Similar and more refined studies in the future have great potential to improve RSL reconstructions in the Mediterranean.

## Supporting information

**S1 File. Benthic foraminiferal abundances in cores BRB-2 and BRB-5, and taxonomic reference list of the most common foraminifera species (> 5%) presented in the text and their ecological preference, i.e., salinity-tolerance ranges.**
(XLSX)

**S2 File. Ostracod abundances in cores BRB-2 and BRB-5, and taxonomic reference list of the most common ostracod species (> 2%) discussed in the text and their ecological preferences, i.e., salinity-tolerance ranges.**
(XLSX)

**S3 File. Total concentrations of trace elements from samples of cores BRB-1, BRB-2 and BRB-5.**
(XLSX)

## Acknowledgments

We thank the graphic artist Noga Yoselevich from the Department of Geography and Environmental Studies, University of Haifa, for the figures' design. Special gratitude goes to Jona Petešić, associate for cultural heritage at the Nature Park Telašćica on Dugi Otok, whose experience and local knowledge greatly helped us while surveying the submerged saltpans of Brbinj. We would like to express our gratitude to Petar Crnčan from the Croatian Natural History Museum for his assistance in identifying the macro-fauna on the site. We thank the three anonymous reviewers, who provided constructive comments and suggestions which helped to improve the manuscript.

## Author Contributions

**Conceptualization:** Benny Bechor, Dorit Sivan.

**Formal analysis:** Benny Bechor, Simona Avnaim-Katav, Naomi Porat.

**Investigation:** Benny Bechor, Simona Avnaim-Katav, Steffen Mischke, Slobodan Miko, Ozren Hasan, Maja Grisonic, Barak Herut, Nimer Taha, Naomi Porat.

**Methodology:** Benny Bechor, Naomi Porat, Dorit Sivan.

**Resources:** Benny Bechor.

**Supervision:** Slobodan Miko, Irena Radić Rossi, Dorit Sivan.

**Validation:** Benny Bechor, Steffen Mischke, Ozren Hasan.

**Visualization:** Simona Avnaim-Katav, Maja Grisonic.

**Writing – original draft:** Benny Bechor.

**Writing – review & editing:** Simona Avnaim-Katav, Steffen Mischke, Dorit Sivan.

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
