## [Decision Letter · Decision Letter 0]

14 Apr 2023

PONE-D-23-07685

How can past sea level be evaluated from traces of anthropogenic layers in ancient saltpans?

PLOS ONE

Dear Dr. Bechor,

Thank you for submitting your manuscript to PLOS ONE. After careful consideration, we feel that it has merit but does not fully meet PLOS ONE’s publication criteria as it currently stands. Therefore, we invite you to submit a revised version of the manuscript that addresses the points raised during the review process.

We look forward to receiving your revised manuscript.

Kind regards,

Fabrizio Frontalini

Academic Editor

PLOS ONE

Journal Requirements:

2. In your manuscript, please provide additional information regarding the specimens used in your study. Ensure that you have reported specimen numbers and complete repository information, including museum name and geographic location. 

For more information on PLOS ONE's requirements for paleontology and archeology research, see https://journals.plos.org/plosone/s/submission-guidelines#loc-paleontology-and-archaeology-research.

   "This study at the Department of Maritime Civilizations, School of Archaeology and Maritime Cultures, University of Haifa, was funded by a Sir Maurice and Lady Irene Hatter Research grant for Maritime Studies for which we are thankful. The research was partly funded through the Croatian Science Foundation grant agreement IP-04-2019-8505 QMAD.We thank the graphic artist Noga Yoselevich from the Department of Geography and Environmental Studies, University of Haifa, for the figures’ design. Special gratitude goes to Jona Petešić, associate for cultural heritage at the Nature Park Telašćica on Dugi Otok, whose 

experience and local knowledge greatly helped us while surveying the submerged saltpans of Brbinj. We would like to express our gratitude to Petar Crnčan from the Croatian Natural History Museum for his assistance in identifying the macro-fauna on the site."

a. You may seek permission from the original copyright holder of Figure(s) [#] to publish the content specifically under the CC BY 4.0 license.  

Additional Editor Comments:

I have now received the comments of three external reviewers raising major concerns on your Ms, particularly Reviewer 1 who rejected it. Reviewer 1 recognizes the value of the Ms but also stresses its limitations and how it can make a significant contribution to sea level studies. The reviewer suggests a re-focused of the Ms and provides some good points to do so. Reviewer 2 also acknowledges the outcomes but finds results not entirely conclusive and some parts unclear. Reviewer 3 highlights the potential of the Ms and considers it novel. The same reviewer suggests to provide a more robust paleoenvironmental interpretation of core units, to better constrain the ecology of species and include a part with the data limitation. On the basis of their comments, I am very sorry to not accept the Ms in the present form and reject the Ms but I strongly invite the authors to benefit of reviewers’ comments and encourage its resubmission.

Reviewers' comments:

Reviewer's Responses to Questions

**Comments to the Author**

1. Is the manuscript technically sound, and do the data support the conclusions?

Reviewer #1: Partly

Reviewer #2: Partly

Reviewer #3: Partly

2. Has the statistical analysis been performed appropriately and rigorously? 

Reviewer #1: N/A

Reviewer #2: Yes

Reviewer #3: N/A

3. Have the authors made all data underlying the findings in their manuscript fully available?

Reviewer #1: Yes

Reviewer #2: Yes

Reviewer #3: Yes

4. Is the manuscript presented in an intelligible fashion and written in standard English?

Reviewer #1: Yes

Reviewer #2: Yes

Reviewer #3: Yes

5. Review Comments to the Author

Reviewer #1: Manuscript #: PONE-D-23-07685

Title: How can past sea level be evaluated from traces of anthropogenic layers in ancient saltpans?

Authors: Benny Bechor; Simona Avnaim-Katav; Steffen Mischke; Slobodan Miko; Ozren Hasan; Maja Grisonic; Irena Radić Rossi; Barak Herut; Nimer Taha; Naomi Porat; Dorit Sivan

Article type: Research Article

Dear Dr Frontalini,

Thank you so much for considering me to review this manuscript (details provided above). This manuscript reads quite well and was also very interesting. As and introduction, I work on coastal system and environmental changes on those systems using chronological, sedimentological, geochemical and micropaleontological proxies. This manuscript sounded really interesting to me, I must say. I would say that the research is very valuable and worth publishing but, and here is the “but”, I do not see how this work responds to the goal of using these anthropogenic remains for sea-level studies. Much less at the time scales that the authors identified. I am sorry to say but I struggle to see how this study make a significant contribution to sea level studies. Few notes on why I say this. The authors did not create sea level index points (SLIPs) and as in fact they mention limiting points. In fact, these locations need to be fully subtidal to allow tides to properly flood them and allow us to, more or less successfully, create a relationship between the depositional environment and the tidal frame. If they are only flooded partially, the problem we would encounter would be that the hydrodynamics are quite complex and we cannot reconstruct the changes in the tidal frame locally. At larger time scales, this could not be such a great problem but here we are talking about contributions for the last 1000 years. To add to this, the region has significant tectonic activity with vertical land movements that exceed 2 mm/yr locally (based on GPS data). Combining isostatic and tectonic vertical land movement makes it more complicated to deconvolve the factors controlling the relative sea-level changes, which, in turn, requires higher precision SLIPs. In all fairness, I kind of struggle to see how the hydrodynamics works here although I know that the authors refer to other works but that requires from the reader/reviewer to do our own research. To finalize this section, I think that the analysis of the tectonic activity and GIA data along with other SLIPs is quite brief and would need further detail (if that is the goal of the paper). For instance, are all the SLIPs in Fig 11 located with enough proximity to each other that vertical land movements would allow direct comparison? As I said before, I think this is an interesting manuscript, but I really struggle with the potential use for sea-level studies.

That said, it really reminds me of works done by Cearreta and collaborators in the North of Spain in reclaimed coastal areas. I think the authors could refocus the manuscript that way. If they do so, I think that the micropaleontological and geochemical analysis would need to be a bit more detailed. I also think that the disagreement between the historical data and the older OSL sample need to be addressed more in depth. Considering that this is one single sample, it could be considered what could have affected the result, such as sediment mixing.

I am sorry that I am not being more supportive and I hope that the authors can revisit the manuscript and get the research published.

Reviewer #2: This is a very interesting manuscript, but despite the abundance of data it does not provide a conclusive interpretation. It is a manuscript based on sedimentological and paleontological (foraminifera and ostracods) description of three sediment cores. Unfortunately, the discussion is not based on interpretations of the data obtained from the cores.

Therefore, the foraminifera, geochemical data, and ostracods are an important source of information that need more attention and better interpolation in the text. The interpretation of the conditions in the saltpans is speculative.

The whole story is based on representatives of the genus Ammonia. For some species, such as A. veneta or A. parkinsoniana, the fact that they are euryhaline, for authors, is very important and has been emphasized several times to obtain more saline conditions. But euryhaline species in themselves are characterized by covering a wide range of salinity. To be consistent with their interpretation, they referred to work in which species were abundant in saline environments and failed to say anything about ammonias from the eastern Adriatic and their distribution in brackish water (the paragraph needs to be better referenced).

Why were changes in foraminifera composition not used to show changes in saline areas (Supplementary data 1)? There is no biodiversity or ecological indices (studying of aliquots of 50 specimens?) to suggest that conditions have become more severe.

Why did the authors use the term dead foraminifera? What types of foraminifera can we find in historical material?

The same problem that data from ostracod assemblages are not considered.

The geochemical data as tracers for identification of a terrestrial: line 455 and man made reddish layer and line 471… potentially resulted from anthropogenic activities…

Specific remarks

Brunovic et al. have studied cores from the island of Cres (northern Adriatic not Dalmatian coast)

WORMS for taxonomic identification, and not Cimerman & Langer?

Reviewer #3: The manuscript is very interesting in approach and it furnishes multi-proxy data (sedimentological, paleontological and geochemical data) from substantially unexplored, Mediterranean sedimentary successions: man-made intertidal saltpans. I’ve also appreciated the use of OSL dating, which proved to be an efficient chronological method within these depositional contexts, deprived of organic remains.

The Authors highlight the novelty of this study mainly in terms of RSL reconstructions. I substantially agree, however I think that paleoenvironmental interpretations (upon which RSL data are based) needs to be improved.

My major comments are as follows:

• A solid paleoenvironmental interpretation of core units is necessary. Each unit is described (4. Results) but not interpreted, with the exception of the paved layer and the so called “euryhaline interval” (first part of the discussion). I suggest to better discuss all the paleoenvironmental phases identified within the studied succession (5.1.) before presenting the RSL inferences (5.2.): pre-saltworks interval; saltworks interval (including the paved layer and the “euryhaline layer”) and post-abandonment interval highlighting key features and differences.

• The Authors stated that they used salinity ecological groups, however in the main text they reported “coastal/lagoonal taxa” about foraminifers (e.g., line 311) and they referred to coastal taxa in Figures 4, 7. I think that this is confusing and the merging of different salinity groups determines a loss of information. Moreover, I wonder which is the meaning of the taxa highlighted in green in the supplementary data 1.

• The counted shells/valves are generally low, I suggest to take into consideration/discuss this issue and to highlight the smallest samples (<50).

• Lines 517-518: from which core sample the OSL age of 1215±175 CE has been obtained? This RSL index point is not clear to me.

Kind regards

6. PLOS authors have the option to publish the peer review history of their article (what does this mean?). If published, this will include your full peer review and any attached files.

Reviewer #1: No

Reviewer #2: No

Reviewer #3: No

---

## [Author Response · Author response to Decision Letter 0]

16 May 2023

1. Answer: The revised Supplementary data 1 & 2 contain the required information regarding the specimens used in our study. We added information about the location where the collected specimens are housed for potential future analyses in the Materials and Methods chapter as follows: "The reported specimens are stored for potential future analysis in the Croatian Geological Survey (HGI), Zagreb".

No permits were required for the described study, which complied with all relevant regulations

2. Answer: Figure 1a-c was replaced, its source now is: "public domain ESRI, Maxar, Earthstar geographics, and the GIS User Community". Figure 1d is a Digital Surface Model of the site, generated by the author and first published by Bechor et al. (2020). The figure caption has been changed in the revised manuscript accordingly, also stated: "The figures are similar but not identical to the original images and is therefore for illustrative purposes only".

---

## [Decision Letter · Decision Letter 1]

20 Jun 2023

How can past sea level be evaluated from traces of anthropogenic layers in ancient saltpans?

PONE-D-23-07685R1

Dear Dr. Bechor,

We’re pleased to inform you that your manuscript has been judged scientifically suitable for publication and will be formally accepted for publication once it meets all outstanding technical requirements.

Kind regards,

Fabrizio Frontalini

Academic Editor

PLOS ONE

Additional Editor Comments (optional):

Reviewers' comments:

Reviewer's Responses to Questions

**Comments to the Author**

1. If the authors have adequately addressed your comments raised in a previous round of review and you feel that this manuscript is now acceptable for publication, you may indicate that here to bypass the “Comments to the Author” section, enter your conflict of interest statement in the “Confidential to Editor” section, and submit your "Accept" recommendation.

Reviewer #3: All comments have been addressed

2. Is the manuscript technically sound, and do the data support the conclusions?

Reviewer #3: Yes

3. Has the statistical analysis been performed appropriately and rigorously? 

Reviewer #3: N/A

4. Have the authors made all data underlying the findings in their manuscript fully available?

Reviewer #3: Yes

5. Is the manuscript presented in an intelligible fashion and written in standard English?

Reviewer #3: Yes

6. Review Comments to the Author

Reviewer #3: I read the revised version of the manuscript written by Bechor and colleagues, and I am satisfied with the performed modifications and responses to my comments. I think this new version of the manuscript is clearly improved and it is ready for final acceptance and publication.

kind regards

7. PLOS authors have the option to publish the peer review history of their article (what does this mean?). If published, this will include your full peer review and any attached files.

Reviewer #3: No

---

## [Editor Report · Acceptance letter]

22 Jun 2023

PONE-D-23-07685R1 

*How can past sea level be evaluated from traces of anthropogenic layers in ancient saltpans?*

Dear Dr. Bechor:

I'm pleased to inform you that your manuscript has been deemed suitable for publication in PLOS ONE. Congratulations! Your manuscript is now with our production department. 

Kind regards, 

on behalf of

Dr. Fabrizio Frontalini 

Academic Editor

PLOS ONE